# Correlation of Gravity and Magnetic Field Changes Preceding Strong Earthquakes in Yunnan Province

Dong Liu [1,2,*], Jiancheng Li [1,3,*], Zhe Ni [2], Yufei Zhao [2], Qiuyue Zheng [2] and Bin Du [1]

[1] School of Geodesy and Geomatics, Wuhan University, Wuhan 430079, China; bdu@whu.edu.cn
[2] Earthquake Administration of Yunnan Province, Kunming 650224, China; a_zhe2006@126.com (Z.N.); zyf18788542726@163.com (Y.Z.); zhengqymoon@163.com (Q.Z.)
[3] MOE Key Laboratory of Geospace Environment and Geodesy, Wuhan University, Wuhan 430079, China
[*] Correspondence: doumo@whu.edu.cn (D.L.); jcli@sgg.whu.edu.cn (J.L.)

**Featured Application: The analysis of the fusion characteristics of gravity and magnetic fields has a certain significance for the prediction of strong earthquakes.**

**Abstract:** The annual variation trend of the gravity and lithospheric magnetic field for adjacent periods are analyzed by using the observation of rover gravity and geomagnetic fields in Yunnan from 2011 to 2021, which tend to be consistent every year during the seismogenic process of a strong earthquake. Thus, this study normalizes the annual value of the adjacent periods for the gravity and lithospheric magnetic field. The normalized values are converted into two classifications that can be compared within [−1,1]. In Yunnan Province, a grid of $0.1° \times 0.1°$ was used to compare the data correlation between the variation of gravity and the variation in the lithospheric magnetic field at the same location. The results are as follows. First, the variation trend of the gravity field and total magnetic field tend to be synchronous year to year in strong earthquake years. The range of consistency increases gradually with the approach of the earthquake year reaching its maximum one year before the earthquake. Throughout the region, the overlap number of normalized annual variations in gravity and magnetic field reaches its maximum, and the peak difference of kernel density curve reaches its minimum. Second, the correlation coefficient of the annual variation in the gravity and magnetic field increases year to year during the development of a strong earthquake within a smaller region surrounding the event. The maximum appears one year before the earthquake, and after the earthquake, the correlation decreases. The analysis of gravity and magnetic fusion characteristics can be employed for the prediction of strong earthquakes.

**Keywords:** gravity and magnetic field fusion; strong earthquake; normalization; nuclear density curve; correlation coefficient

## 1. Introduction

Studies have shown that the mobile gravity monitoring network in Yunnan and surrounding areas can identify gravity changes related to moderate and strong earthquakes [1,2]. The changes in the gravity field are closely related to the changes in the precursors of moderate and strong earthquakes in this region, near the transition zone of positive and negative changes in the field and the high gradient zone [3–5]. In addition, domestic experts and scholars also made mid-term and long-term predictions for the strong earthquake of Yao'an $M_s6.0$ in July 2009 and Jinggu $M_s6.6$ in October 2014, which achieved good prediction results [6,7]. In the middle of the 20th century, scientists in some countries, such as the United States, Russia, Japan, Kazakhstan, and China, carried out related studies on the relationship between geomagnetic changes and earthquakes [8–11]. After the Wenchuan earthquake, the geomagnetic survey technology team of the China Earthquake Administration carried out annual geomagnetic field observations in Great North China,

the north–south seismic belt and the north–south Tianshan Mountains. The total intensity gradient zone in the obtained lithosphere magnetic field also has a good seismic reflection relationship with the epicenter location [12–14].

From 2011 to 2021, there were four earthquakes of $M_s$6.0 and above in Yunnan Province: Yingjiang $M_s$6.1 (25.0° N, 97.8° E) on 30 May 2014, Ludian $M_s$6.5 (27.1° N, 103.3° E) on 3 August 2014, Jinggu $M_s$6.6 (23.4° N, 100.5° E) on 7 October, and Yangbi $M_s$6.4 (25.7° N, 99.9° E) on 21 May 2021. In the period of earthquake preparation and seismogenesis, both the gravity field and the total magnetic strength change greatly, and the range of positive and negative anomalies tends to be consistent year to year as the seismogenesis time approaches. In Figure 1, the spatial distribution of the annual difference in gravity and magnetic anomaly fields are similar for each of the two years that precede strong earthquakes in the Yunnan area.

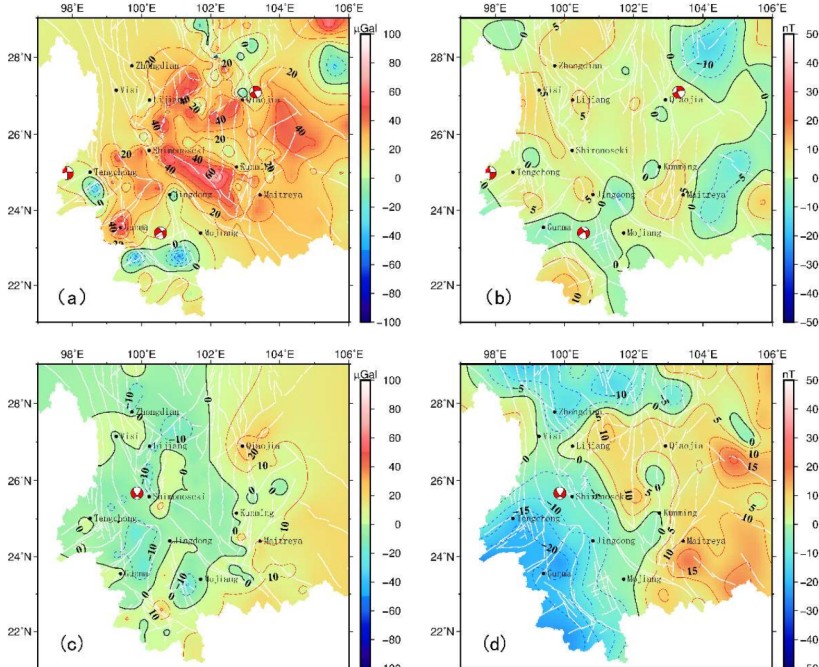

**Figure 1.** The spatial distribution of the annual difference in gravity and magnetic anomaly fields for each of the two years that precede strong earthquakes in the Yunnan area. (**a**) Annual changes in gravity from 2013 to 2014; (**b**) Annual changes in total geomagnetism from 2013 to 2014; (**c**) Annual changes in gravity from 2019 to 2020; (**d**) Annual changes in geomagnetism from 2019 to 2020. Epicenter and focal mechanism are shown for earthquakes that occurred within a year of the respective time period end.

In this paper, the annual variations in gravity and the magnetic field were normalized to scalar values in the range of [−1,1] by comparing and analyzing the gradual consistency of the annual variations of gravity and magnetic field in the process of seismogenic earthquake, and then the whole measurement area was processed by grids with a resolution of 0.1° × 0.1°. The number of requalified nodes with each variation trend in the interval [−1,1] was obtained, and the correlation coefficients of the two nodes were compared in the seismogenic process. The seismogenic characteristics of strong earthquakes were analyzed from the magnitude values after fusion, and the relationship between the fusion physical field and the seismogenic process of strong earthquakes was extracted.

## 2. Data Processing and Accuracy

### 2.1. Gravity and Geomagnetic Field Data Acquisition Instrument and Research Area

There are 249 gravity measurement points in Yunnan Province and its surrounding areas, including nine absolute gravity points. The absolute observation adopts the FG-

5 absolute gravimeter, and the observation accuracy is $2 \times 10^{-8}$ m·s$^{-2}$. The relative observation instrument is CG-5. The relative gravimeter has an observation accuracy of $10 \times 10^{-8}$ m·s$^{-2}$; there are 130 geomagnetic vector measurement points, of which the GSM-19T proton precession magnetometer (PPM) is used to observe the total geomagnetic intensity, the instrument resolution is 0.1 nT, and the observation accuracy is 0.5 nT.

The research area of this paper is 21~29° N, 97~106° E, and the research objects are mainly the observation data of gravity and geomagnetic measurement points in this area. The distribution of measuring points and measuring lines is shown in Figure 2.

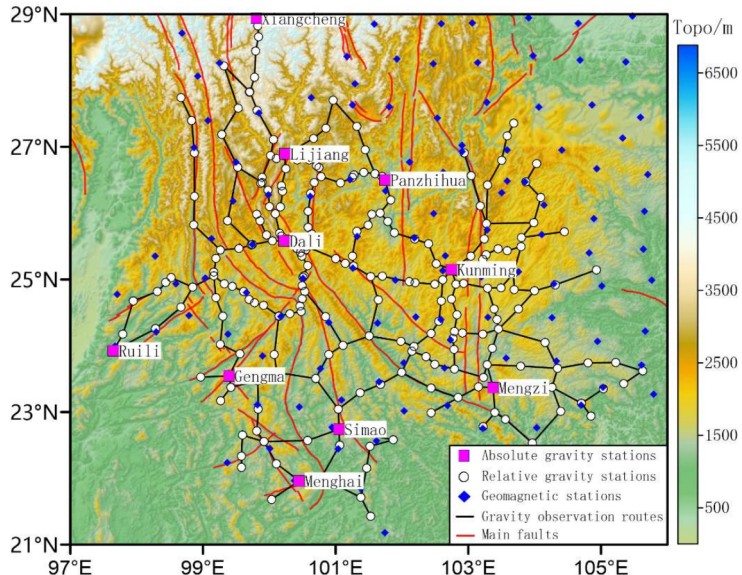

**Figure 2.** Distribution map of gravity survey points, geomagnetic survey points, and faults in Yunnan Province.

## 2.2. Data of Gravity and Geomagnetic Total Intensity

### 2.2.1. Gravity Data Processing and Accuracy

Every year, the Yunnan Seismological Bureau carries out two relative gravity observations in the whole province and conducts joint surveys with 10 absolute gravity points in the survey area. The gravity value of each point is obtained through adjustment calculation. In the adjustment calculation of gravity data, the observation data are first adjusted by a free network, and the observation accuracy of each instrument is preliminarily understood. Second, according to the observation accuracy of each instrument, the weight of each instrument is determined reasonably. Finally, absolute gravity control was used to carry out classical adjustment for the pretreatment results [15]. The accuracy of the adjustment calculation is shown in Table 1.

In Table 1, the average accuracy of the 20 data points from 2011 to 2021 is (6.6~10.5) $\times 10^{-8}$ m·s$^{-2}$, the average point value accuracy is $8.5 \times 10^{-8}$ m·s$^{-2}$, and the difference between the posterior median error and the prior median error is small, reflecting the reliable quality of gravity observation data, which can be used to reflect the space–time dynamic evolution process of the regional gravity field.

**Table 1.** Accuracy of gravity data processing results from March 2011 to September 2021.

| Data Group | Mean of Value Precision /$10^{-8}$ m·s$^{-2}$ | Posterior Error /$10^{-8}$ m·s$^{-2}$ | Posterior In-Error Minus Prior-In-Error /$10^{-8}$ m·s$^{-2}$ |
|---|---|---|---|
| March 2011 | 8.6 | 8.3 | 0 |
| September 2011 | 7.5 | 9.7 | 0 |
| March 2012 | 6.6 | 6.8 | 0 |
| September 2012 | 10.5 | 6.4 | 0 |
| March 2013 | 8.9 | 4.7 | 0 |
| September 2013 | 10.4 | 8.2 | 0 |
| March 2014 | 9.1 | 5.9 | 0.6 |
| September 2014 | 7.3 | 6.5 | 0 |
| March 2015 | 7.4 | 6.3 | 0.6 |
| September 2015 | 8.4 | 7.0 | 0 |
| March 2016 | 8.3 | 8.6 | 0.4 |
| September 2016 | 8.3 | 8.5 | 0.5 |
| March 2017 | 8.8 | 8.6 | 0.4 |
| September 2017 | 9.1 | 8.8 | 0.2 |
| March 2018 | 8.1 | 9.6 | 0.4 |
| September 2018 | 10 | 11.1 | −1.1 |
| March 2019 | 8.5 | 9.9 | 0.1 |
| September 2019 | 7.9 | 11.5 | −1.5 |
| March 2020 | 6.9 | 9.9 | 0.1 |
| September 2020 | 9.3 | 10 | 0 |
| March 2021 | 10.5 | 11.5 | −1.5 |
| September 2021 | 7.6 | 8.0 | −0.4 |

### 2.2.2. Geomagnetic Data Processing and Accuracy

The instrument used to measure the total strength of geomagnetism was the GSM-19T PPM. According to the principle of nuclear magnetic resonance, the instrument reflects the total strength of the magnetic field through the frequency of the signal. It has good stability and is not affected by temperature, humidity, etc., and the observation accuracy is high, the operation is simple, the instrument is easy to carry, and it has the characteristics of digitization of data collection and automation of precise measurement.

To acquire the total magnetic field measurements, a GSM-19T (GEM Corporation, Canada) PPM with a sensitivity of 0.15 nT @ 1 Hz, a resolution of 0.01 nT and an absolute accuracy of ±0.2 nT were employed.

The daily variation in the nearest station substitution method and the long-term variation correction of the natural orthogonal quantity model were used to obtain the annual variation of the lithospheric magnetic field. Diurnal variation is mainly used to eliminate regular diurnal variation and other exogenous fields in observed data, and long-term variation is aimed at obtaining the annual variation in the lithospheric magnetic field [16].

The annual variation in the lithospheric magnetic field is calculated by Formula (1):

$$\Delta F_{Lith\_T_2 - T_1} = (F_{Int\_T_2} - F_{Int\_T_1}) - (F_{Main\_T_2} - F_{Main\_T_1}) \tag{1}$$

In the calculation formula, $\Delta F_{Lith\_T_2 - T_1}$ is the difference between the lithospheric magnetic field from $T_2$ to $T_1$ of a certain geomagnetic element, $F_{Int\_T_i}$ is the endogenous magnetic field of the geomagnetic element (the result of daily variation), and $F_{Main\_T_i}$ is the main magnetic field (the long-term regular change).

### 2.3. Calculation Method

First, the gravity variation values and geomagnetic total intensity values of the same year were grid treated. Second, the scalar data in the range of [−1,1] were obtained by normalization of their respective change values. Finally, the quantity distribution of gravity variation value and total intensity value in the same year were summarized, the kernel density curve of the quantity distribution was generated, and the correlation coefficient between gravity variation and geomagnetic total intensity change falling within the same year was calculated.

### 2.3.1. Grid Interpolation

The Yunnan area was gridded according to the longitude difference of $0.1°$ and the latitude difference of $0.1°$, and kriging interpolation was performed on the grid points to calculate the gravity change value and the total geomagnetism value of the grid points.

$$\hat{z}_0 = \sum_{i=1}^{n} \lambda_i z_i \tag{2}$$

In Formula (2), $\hat{z}_0$ is the estimated value of the measuring point $(B_0, L_0)$. That is, $\hat{z}_0 = z(B_0, L_0)$, where $\lambda_i$ is the weight coefficient. A set of optimal coefficients can meet the minimum difference between the estimated value at measurement point $\hat{z}_0$ and the true value $z$.

### 2.3.2. Normalization Processing and Correlation Coefficient Calculation

The grid gravity change value and the total geomagnetism value are normalized and converted into a dimensionless scalar with a value range of $[-1,1]$. To avoid changing the positive or negative sign of the change value, the following method was used in the normalization process.

$$X_{norm} = \frac{X}{|X|_{\max}} \tag{3}$$

In Formula (3), $X$ is the change value and $|X|_{\max}$ is the maximum absolute value of the interannual change value.

To accurately show the relationship between the gravity change value and the total geomagnetism value, the normalized result of the two is used as a variable, and the linear correlation coefficient between the two was calculated by Formula (4).

$$r(X_{Gra}, X_{Geo}) = \frac{Cov(X_{Gra}, X_{Geo})}{\sqrt{Var[X_{Gra}]Var[X_{Geo}]}} \tag{4}$$

In the formula, $r$ is the correlation coefficient, $X_{Gra}$ is the normalized gravity value, $X_{Geo}$ is the normalized total geomagnetic intensity, $Cov(X_{Gra}, X_{Geo})$ is the covariance of the gravity value and the total strength of the magnetic field, $Var[X_{Gra}]$ is the variance of the gravity value, and $Var[X_{Geo}]$ is the total geomagnetism variance.

## 3. Results

### 3.1. Histogram and Nuclear Density Plot of Consistent Data Quantity in the Change Interval

The number distribution of normalized gravity change and total intensity change is in the interval $[-1,1]$. The overlap of the two intervals represents the number of grid points with the same normalized change in the entire area. The nuclear density curve is also used to visually indicate the overlap of the two intervals. The area enclosed by the nuclear density curve is 1. The larger the upper limit of the vertical axis of the nuclear density graph is, the more concentrated the data change range is near 0, and the more overlapped the area is. More means that the more grid points with the same amount of change, the more consistent the change trend is. The histogram and nuclear density diagram of the uniform number of grid point changes in the normalized gravity field and magnetic field from 2011 to 2021 are shown in Figure 3.

As shown in the above figures, from 2011 to 2012 and from 2012 to 2013, the degree of coincidence for the same interval of gravity and magnetic field changes was low; the difference in the kernel density map was large. From 2013 to 2014, the overlap interval between the gravity and total intensity changes was approximately $-0.2$ to $0.6$; the peak value on the gravity change curve in the kernel density graph was slightly greater than 2.0, the peak value of the total intensity change was slightly less than 2.0, and the difference between them was approximately 0.25. Three earthquakes with magnitudes of 6.0 or greater occurred in one year from 2014 to 2015. The overlap interval between the change in the

gravity and the change in the total intensity was approximately −0.75; the peak value on the gravity change curve in the kernel density graph was approximately 2.8, and the peak value of the total intensity change was approximately 1.6. The difference between them was approximately 1.2. From 2015 to 2016, 2016 to 2017, 2017 to 2018, and 2018 to 2019, the gravity magnetic coincidence degree was the best from 2016 to 2017, but the kernel density difference was large. Six earthquakes with magnitudes of 5.0 and greater occurred during this period, and no strong earthquakes occurred. From 2019 to 2020, the coincident range of gravity variation and total intensity variation was approximately −0.75 to 0.75. In the kernel density diagram, the peak value of the gravity variation curve was approximately 1.1, and the peak value of the total intensity variation was approximately 1.2, with a difference of approximately 0.1. From 2020 to 2021, an earthquake with a magnitude of 6.4 occurred.

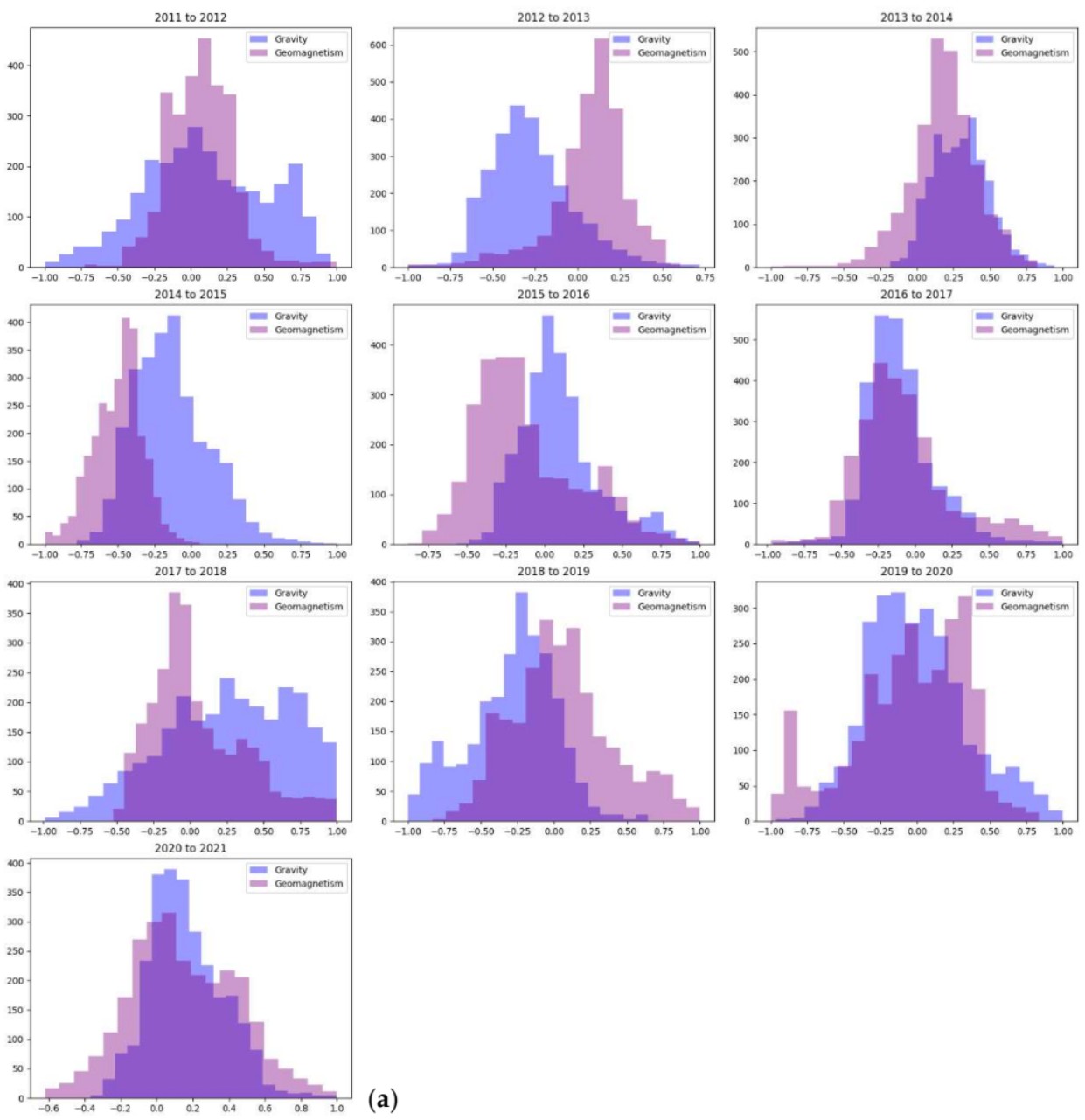

**Figure 3.** *Cont.*

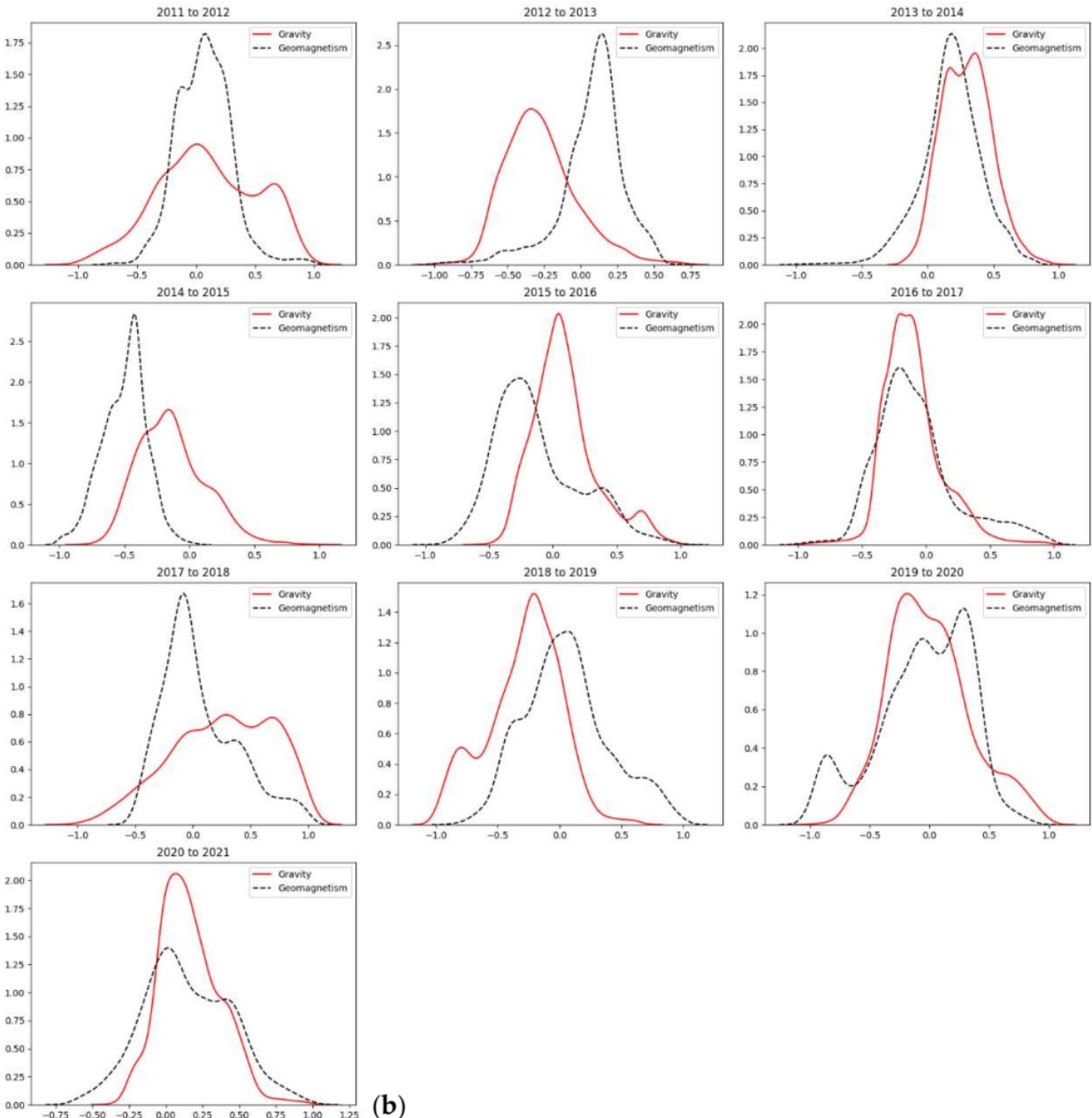

**Figure 3.** The histogram and nuclear density map of the uniform number of grid point changes in the gravity and magnetic field after normalization from 2011 to 2021. (**a**) Histogram of uniform number of grid points of gravity and magnetic field after normalization (the ordinate represents the quantity and the abscissa represents the normalized interval value); (**b**) The normalized gravity and magnetic field grid points change uniform number nuclear density map (the ordinate represents the kernel density value, and the abscissa represents the normalized interval range value).

### 3.2. Calculation of the Correlation Coefficient of Gravity and Magnetic Field Data

According to the analysis of gravity anomaly indicators, the range of gravity anomalies for earthquakes with magnitudes $M_s6.0$ to $7.0$ is 220 to 350 km [17]. The data gridding and normalization of the seismic anomaly area were performed on Yingjiang $M_s6.1$, Ludian $M_s6.5$, Jinggu $M_s6.6$, and Yangbi $M_s6.4$, and the correlation analysis of the annual change data of the gravity and magnetic field was performed. According to Formula (4), the correlation coefficient of annual variation data of gravity and magnetic field from 2011 to 2021 is shown in Figure 4.

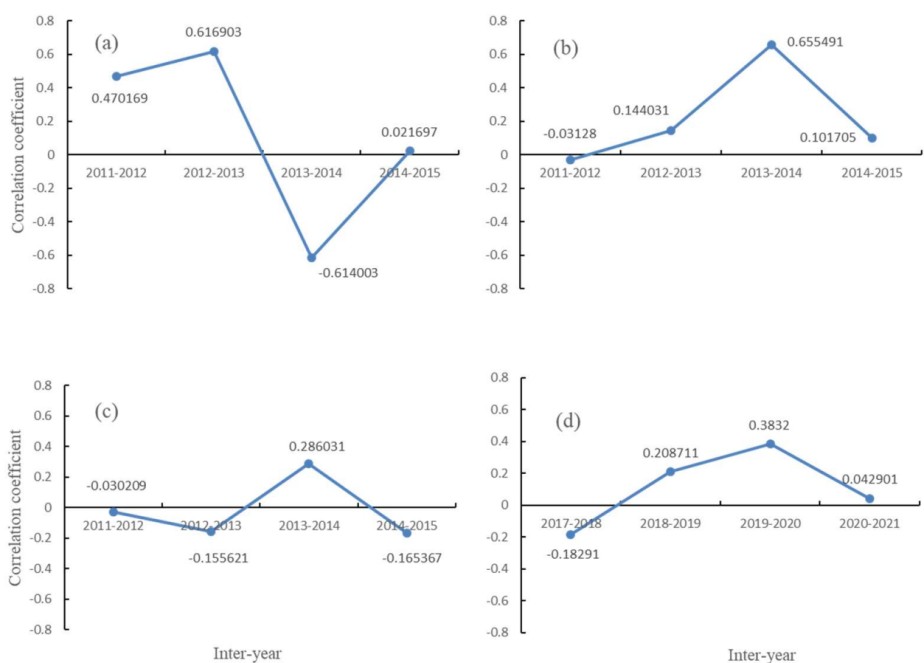

**Figure 4.** The strong earthquakes in Yunnan Province from 2011 to 2021 and the changes in the correlation coefficient of gravity and magnetic field within the three-year earthquake area before the earthquakes. (**a**) Yingjiang $M_s$6.1 earthquake in 2014; (**b**) Ludian $M_s$ 6.5 earthquake in 2014; (**c**) Jinggu $M_s$ 6.6 earthquake in 2014; (**d**) Yangbi $M_s$ 6.4 earthquake in 2021.

In Figure 4a, Yingjiang is located on the China–Myanmar border, surpassing China's border without gravity magnetic field data, and the amount of data used in the calculation of the correlation coefficient is relatively small. The absolute value of the correlation coefficient in the year before and the second year before the earthquake is relatively large. One year before the occurrence of the four strong earthquakes in Figure 4, the correlation coefficient of the annual variation of the gravity and magnetic field in the area covered by the anomaly reached the maximum, and the correlation coefficient decreased sharply in the year of the earthquake.

## 4. Discussion

During the earthquake preparation process of the four strong earthquakes in Yunnan from 2011 to 2021, the trends of gravity and magnetic field changes tend to be consistent in the three years before the earthquake. After the data are normalized, the amount of data overlap increases year to year, and the nuclear density difference decreases year to year. In particular, there is a number of overlaps of the gravity and magnetic field fusion data in the year before the earthquake reaches the maximum, the overlapped data distribution interval is concentrated, and the nuclear density difference is less than 0.25. In the year of the earthquake, the number of coincident data decreases, the distribution interval is scattered, and the nuclear density difference increases. The correlation coefficient of gravity and the magnetic field increases year to year during the seismogenic process and reaches the maximum value one year before the earthquake. The correlation coefficient decreases sharply in the year of the earthquake.

The gravitational field is formed by the combined force of the Earth's gravitational force on an object and the inertial centrifugal force produced by the Earth's rotation. The lithospheric magnetic field is produced by lithospheric magnetic material, and the two fields are unrelated [18–20]. Under normal conditions (constant density and magnetic susceptibility), the gravitational field and the lithospheric magnetic field are both stable field sources. During the seismogenic process of a strong earthquake, along with tectonic movement, the rock is stressed, and the density and magnetic susceptibility change. The gravity value and geomagnetism change the total strength [21,22]. The change in the

tectonic stress on the field source causes the change in the mass distribution of the local area, resulting in the zoning pattern of gravity on the field and the abnormal zone along the structure [23,24]. At the same time, the change in the rock stress causes the magnetic susceptibility to change. This results in a regional anomaly feature of the change in the lithospheric magnetic field [25]. The gravity anomaly in the strong earthquake danger zone is characterized by the occurrence of long-term and large-scale multipoint anomalies in the epicentral area, and the zoning pattern is distinguished along the active structure and the cascade and anomaly zones distributed along the structure in the periphery. The characteristics of mobile geomagnetic anomalies in strong earthquake danger areas are as follows: The annual variability of mobile geomagnetic anomalies near the epicenter is concentrated in a certain spatial distribution. During the six months to one year before the earthquake, there are several abnormal regions of total intensity of geomagnetic points near the epicenter and in specific geological structures [26].

Under the action of plate tectonics, the gravity and magnetic fields gradually tend to be consistent in the process of strong earthquake preparation and weaken or reverse after the earthquake [27,28] The results show that the compression or tension between plates produces large stress flow during tectonic activities [29]. The stress flow propagates in the rock circle [30]. When encountering areas with drastic changes in medium structure and density (such as faults), the propagation channel encounters obstacles, and the stress flow will change the gravity field and geomagnetic field near the epicenter.

Based on the performance of the annual change in the gravity field and the total geomagnetism in the seismogenic process, and combined with the observation data from 2011 to 2021, the two different field sources of the gravity and magnetic field pass through the change value. After normalization, we jointly study the correlation between the two and the characteristics of the gestation process of strong earthquakes as an exploration of the fusion of two field sources to study the characteristics of strong earthquakes. This article has two shortcomings: First, the time of obtaining gravity and geomagnetic data cannot be guaranteed to be consistent. The time of gravity observation and geomagnetic observation in Yunnan Province is not synchronized, and the difference is 1–2 months, so the calculated correlation coefficient is small. Secondly, there are 25 border counties in Yunnan which are, respectively, joined by Myanmar, Laos, and Vietnam. The national border is 4060 km long, but there are no gravity and geomagnetic data in areas beyond the border, resulting in mutation data in the calculation of correlation coefficient between the two.

Multiple field sources are combined to analyze the mechanism of earthquake preparation, and the characteristics of earthquake occurrence are quantitatively analyzed based on indicators such as correlation coefficients. As an exploration of multifield source fusion to analyze the characteristics of strong earthquakes, this study serves for mid-term prediction of strong earthquakes. In this study, the determination of the epicenter location is not introduced. In the future, we will try to jointly study this problem through ground observation technologies such as the Global Navigation Satellite System and InSAR.

## 5. Conclusions

From the perspective of gravity and magnetic field data fusion, this paper analyzes and studies the characteristics and laws between the occurrence of strong earthquakes and the dynamic changes in gravity and magnetic fields in Yunnan since 2011. The main understandings obtained are as follows:

(1) The fusion of the data for gravitational and the lithospheric magnetic field has a good precursor to the seismogenesis of strong earthquakes. During the seismogenic process, the annual change in gravity and its trend in magnetic field strength have a certain correlation. The positive and negative variation of the gravity field tend to be the same for the total magnetic field intensity, both reaching a maximum one year before the earthquake.

(2)    After the gravity and magnetic field fusion data are normalized, the interannual gravity change and the annual trend change in the total magnetic field strength are converted between −1 and 1. The entire survey area can be distinguished in the same way. According to the grid of rate, the coincidence of gravity and magnetic points between −1 and 1 with the same trend can be obtained. In the year before the strong earthquake, the coincidence number reaches its maximum, and the peak difference of the nuclear density curve reaches the minimum.

(3)    The relationship between magnitude and anomalous range, as well as the gravity and magnetic fields, in the area surrounding a strong earthquake were fused. It shows that the correlation coefficients of change all reach their maximum, and the annual change coefficients decrease sharply in the year of the earthquake.

(4)    Based on the results of this paper, it is feasible to conduct a comprehensive analysis of strong earthquakes by the fusion of multiphysics, but the current problem is that no more intuitive characteristic display is found in the judgement of the location of the earthquake. In the future, ground observation technologies such as GNSS and InSar can be integrated to jointly solve the problem of earthquake elements such as epicenters and accurate determination of earthquake occurrence times.

**Author Contributions:** Conceptualization, J.L., Z.N. and D.L.; methodology, D.L.; software, D.L. and Y.Z.; validation, Q.Z.; writing: original draft preparation, D.L.; writing: review and editing, D.L. and Z.N.; visualization, B.D. All authors have read and agreed to the published version of the manuscript.

**Funding:** This project was funded by the Director's Fund of the Institute of Earthquake Research, China Earthquake Administration (No. IS201926302), the 2021 Earthquake Situation Tracking and Orientation Tasks of the Monitoring and Prediction Department of China Earthquake Administration (No. 2021010228), and the Spatial-temporal evolution of the lithospheric magnetic field anomalies and identification of sub-instability in typical earthquake regions study (No. 2018YFC150330504); Kunming Modern Surveying and Mapping Benchmark Construction (Phase II)—Second-Class Gravity Survey (No. J-CH-2021-020).

**Institutional Review Board Statement:** Not applicable.

**Informed Consent Statement:** Informed consent was obtained from all subjects involved in the study.

**Data Availability Statement:** Not applicable.

**Acknowledgments:** We would like to thank the mobile gravity and geomagnetic survey personnel in the Sichuan–Yunnan area for obtaining high-quality data, the expert guidance of the Institute of Earthquake Research of China Earthquake Administration and the Institute of Geophysics of China Earthquake Administration, and the valuable opinions of reviewers.

**Conflicts of Interest:** No conflicts of interest exist in the submission of this manuscript, and the manuscript has been approved by all authors for publication.

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
