# Peer review of "Correlation of Gravity and Magnetic Field Changes Preceding Strong Earthquakes in Yunnan Province"

_applsci, doi:10.3390/app12052658_

Round 1
Reviewer 1 Report
Summary. The manuscript reports a potentially interesting observation that suggests that the pattern/correlation of small changes in gravity and magnetic fields might reflect the build up of stresses during the ~year preceding a large earthquake. However, the presentation is not very compelling. Improvement in the organization and detail of the text and figures could address part of this problem but I am not able to determine whether the content is worthy without additional information.
Presentation: There are a number of incomplete sentences within the text- the English needs to be proof-read and corrected. The style of writing is a somewhat confusing- in several places a statement attempts to introduce or summarize a topic/method and it is not clear whether specific corresponding information will be provided later (or not). I suggest adding a phrase to indicate that details on the topic are 'discussed below in Section ***'. The format of some of the formulas (1, 4is messed up in the PDF. Several references are not complete or are incorrectly formatted.
Specific comments:
Could shorten title: 'Correlation of gravity and magnetic field changes preceding strong earthquakes in Yunnan province'. ??
The Abstract is repetive. Many unnecessary words/phrases are used.
Introduction: need to clarify the sentence that is just above Figure 1. I suggest "The spatial distribution of the annual difference in gravity and magnetic anomaly fields are similar for each of the two years that precede strong earthquakes in the Yunnan area."
Figure 1. add a symbol to show the earthquake epicenters in each panel
Twice a yr sampling is very likely biased, missing shorter-term variation in factors such as solid Earth tides (several mgal) and timing of rainfall within the Fall/Spring seasons. This needs to be mentioned and suggestions for how to address such possible bias in future work could increase the impact of this study.
Also, 'annual variation' is misleading- it is really the difference between Fall and Spring measurements rather than continuous variation throughout the year. The fact that gravity and magnetic measurments are not coincident is reported but the reliability of their comparison is not sufficiently justified. Additional discussion of the expected time scale of variation in each field is needed- is the signal reported here anything more than a combination of spurious climate/tidal/microseismicity related events??
Top PP page 4 is too vague- explain how 'determined reasonably' was actually done: what type/magnitude of weighting was assigned and were individual stations the same every yr or different. If different, why?
Fix Figure 3 caption (put a & b after the general sentence). Add labels 'a' and 'b'.
Section 3.1: eliminate the text that delineates every year's value and difference, since that info is directly evident in Figure 3. Instead, provide a summary paragraph that describes the nature/range of similarity and differences that occur over the period. Combine Sections 3.1 and 3.2 and rename appropriately. Clarify whether (or not) the correlation between gravity and geomagnetic change was computed for only a subset of the data. If so, specify exactly what radius around each epicenter was used (220 or 350 km as suggested by prior studies, or something else?)
I think it may be important to show the correlations for WHOLE period (2011-2021) for each so that the behavior both during and outside the pre/syn-earthquake period is clear. Is the variation during the 3-yr period really unusual in terms of the correlation increase/decrease? Figure 3 suggests that the pattern of temporal increase/decrease is not that straightforward- there are other cases of increase then decrease when no strong earthquake occurred.
Figure 4: I suggest making each panel (a-d) have an upper (correlation) and lower (earthquakes) part. In the lower part, show all earthquakes in the sub-region (time vs magnitude) for the whole period shown
Page 10, top paragraph, second-to-last sentence: must explain what you mean by 'geomagnetic anomalies near the epicentre is con-centrated in a certain spatial distribution'. Describe 'certain' in detail.
IF the correlation results are actual robust, it would be helpful to add some discussion of what magnitude of decrease in correlation might provide a robust indicator for potential earthquake prediction (this is where showing correlation for the full 2011-21 period would be important so that the variability outside a 'strong earthquake gestation process' is quantified).
Conclusions: 1-3 are very repetitive. Rewrite
Author Response
Dear Editors and Reviewers,
Thank you very much for taking your time to review this manuscript. We really appreciate all your comments and suggestions. We have studied these comments carefully and tried our best to revise and improve the manuscript. In the copy of the manuscript with the changes noted, the changes have been highlighted in track changes. In the final revised manuscript, we revised all of comments from reviewers. Our itemized responses to reviewers’ comments are listed below, followed by revisions.

Reviewer 2 Report
The manuscript describes an interesting work in relation to the behavior of gravity and the local magnetic field and its possible relationship with the occurrence of earthquakes of magnitude M => 6 in the studied area. In general, the data analysis is well done, but the discussion and conclusions are not fully substantiated.
The authors state, in their discussion, "During the earthquake preparation process of the four strong earthquakes in Yunnan from 2011 to 2021, the trends of gravity and magnetic field changes tend to be consistent in the three years before the earthquake." In this paragraph some doubts arise, being, in my opinion, the most important that the conditions be explained when the preparation process for a strong earthquake begins and that they must also define what they mean by an earthquake caused by gravity (See first and second line of page 2.)
On the other hand, the authors analyzed the correlation between the fluctuations of the local gravitational and magnetic fields measured in the lithosphere. They stated that these fluctuations are correlated with the earthquakes with a magnitude greater than or equal to 6.
Their analysis is based on the behavior of the distribution functions of the two monitored variables.
In my opinion, it is not clear that these measurements allow to identify a precursor of each earthquake.
In addition, the relationship between seismic dynamics and the mechanisms that lead to the occurrence of the aforementioned earthquakes is not clear.
I consider that the manuscript can be published if seismological arguments are incorporated that allow clarifying the conclusions described by the authors.
Among the suggestions, the authors should describe in a short paragraph expression such as "Moderate and strong earthquakes in Yunnan are mostly caused by gravity". This statement is not clear unless they specify the meaning, from the tectonic point of view, of earthquakes caused by gravity.
In my opinion, the work does not describe enough evidence to establish that the correlation between the fluctuations of the gravitational field and the magnetic field measured are produced by the earthquakes that have occurred, so that the manuscript must be reviewed in their discussion and conclusions.
The equation 1 must be written correctly.
Author Response

(The authors gave the same response as above.)

Round 2
Reviewer 1 Report
The manuscript is improved and, along with the authors responses, I am now convinced that the paper can be ready for publication with minor revision. I still find the results only modestly compelling but I think it is reasonable that the work be made available to the scientific community and these initial indications might well spur additional, more complete future work.
Line (L) 17: change (c/) 'the same' to (/) 'each'
L20: c/'approaching'/'approach'
L21: insert 'Throughout the region' before ' the overlap...'
L24: delete 'for the anomaly area' and insert 'within a smaller region surrounding the event' after 'earthquake'
L25: insert 'the correlation decreases' after 'earthquake'
L34-35: this sentence is incomplete- reword
L 52: c/'In Figure 1, The'/'In Figure 1, the'
L58: delete 'are similar'
L61: c/'Variety'/'. Epicenter and focal mechanism are shown for earthquakes that occurred within a year of the respective time period end.
L80: c/'nt'/'nT' (twice)
L92: c/'preliminary'/'preliminarily'
L159: c/'change in'/'change fall within
L166: insert 'is' after 'trend'
L184: c/'de-gree'/'degree'
L215: c/'earthquake'/'earthquakes'
L251: c/'result is'/'results in'
L264: c/'propa-gates'/'propagates'
L266: c/'and The'/'and the/
L274-275: redundant, delete 'article also has certain shortcomings; that is, the time of the obtained gravity and geomagnetic data cannot be guaranteed to be consistent. This'
L278: delete 'in months'
L299: c/'total intensity reaching its'/'total magnetic field intensity, both reaching a'
L308: c/'anomaly area covered'/'area surrounding a strong earthquake
L310: insert new paragraph before (4)
Author Response
Dear Editors and Reviewers,
Thank you very much for taking your time to review this revision. We really appreciate all your comments and suggestions. We have studied these comments carefully and tried our best to revise and improve the manuscript. In the copy of the manuscript with the changes noted, the changes have been highlighted in track changes. In the final revised manuscript, we revised all of comments from reviewers. Our itemized responses to reviewers’ comments are listed below, followed by revisions.

Reviewer 2 Report
I read the manuscript carefully. In my opinion, the research work is well done, but there are some issues that I think need to be taken into account in order to clarify some ideas.
In order to publish the paper the authors must to attend the following recommendations.
1. On page 2, lines 1 and 2, the authors wrote "Moderate and strong earthquakes in Yunnan are mainly caused by gravity." This phrase seems to indicate that gravity is the source of earthquakes in Yunnan. Authors must explain whether the source of earthquakes is gravity or not. I think the authors have to explain the relationship between the gravity field and the tectonic activity in the study area.
2. In the second paragraph, the authors wrote "In the period of earthquake preparation and seismogenesis, both the gravity field and the total magnetic force change a lot." How can authors know when preparation periods begin?
3. In figure 1 I recommend identifying, with a mark on the graph, the epicenters of each earthquake analyzed.
4. Formula (1) must be corrected.
5. To better understand the use of formula (1), the interval between T1 and T2 means days, months or how long?
6. Formula (4) must be corrected.
7. Figure 3 represents the distribution function of both analyzed parameters. I have a question Why did the authors calculate the correlation using Formula (4)? My question is because the distribution functions suggest that the data sets are not stationary.
I think the mutual information function cloud would be better to estimate the correlation.
8. Finally, I believe that the authors should establish their results with the tectonic properties of the 4 earthquakes studied.
Author Response

(The authors gave the same response as above.)
